# Coronavirus Disease 2019 (COVID-19) Reinfection Rates in Malawi: A Possible Tool to Guide Vaccine Prioritisation and Immunisation Policies

**DOI:** 10.3390/vaccines11071185

**Published:** 2023-06-30

**Authors:** Master R. O. Chisale, Frank Watson Sinyiza, Paul Uchizi Kaseka, Chikondi Sharon Chimbatata, Balwani Chingatichifwe Mbakaya, Tsung-Shu Joseph Wu, Billy Wilson Nyambalo, Annie Chauma-Mwale, Ben Chilima, Kwong-Leung Joseph Yu, Alfred Bornwell Kayira

**Affiliations:** 1Faculty of Sciences, Technology and Innovations, Biological Sciences, Mzuzu University, P/Bag 201 Luwinga, Mzuzu, Malawi; 2Research Department, Luke International, Mzuzu P.O. Box 1088, Malawi; 3Mzuzu Central Hospital, Ministry of Health, P/Bag 209 Luwinga, Mzuzu, Malawi; 4Department of Public Health, University of Livingstonia, Mzuzu 00265, Malawi; 5Overseas Department, Pingtung Christian Hospital, No. 60, Da-lien Rd., Pingtung City 900, Taiwan; 6Public Health Institute of Malawi, Ministry of Health, Lilongwe 00265, Malawi

**Keywords:** reinfection, COVID-19, vaccinations, immunizations, SARS-CoV-2, immune response

## Abstract

As the fight against the COVID-19 pandemic continues, reports indicate that the global vaccination rate is still far below the target. Understanding the levels of reinfection may help refocus and inform policymakers on vaccination. This retrospective study in Malawi included individuals and patients who tested for COVID-19 infections via reverse transcriptase polymerase chain reaction (rt-PCR) from the data at the Public Health Institute of Malawi (PHIM). We included all data in the national line list from April 2020 to March 2022. Upon review of 47,032 records, 45,486 were included with a reported 82 (0.18) reinfection representing a rate of 0.55 (95% CI: 0.44–0.68) per 100,000 person-days of follow-up. Most reinfections occurred in the first 90 to 200 days following the initial infection, and the median time to reinfection was 175 days (IQR: 150–314), with a range of 90–563 days. The risk of reinfection was highest in the immediate 3 to 6 months following the initial infection and declined substantially after that, and age demonstrated a significant association with reinfection. Estimating the burden of SARS-CoV-2 reinfections, a specific endurance of the immunity naturally gained, and the role played by risk factors in reinfections is relevant for identifying strategies to prioritise vaccination.

## 1. Introduction

As of 6 April 2023, an estimated 12 million Africans and more than 680 million individuals worldwide were infected with Severe Acute Respiratory Syndrome Coronavirus 2 (SARS-CoV-2). Furthermore, SARS-CoV-2 has claimed 258,670 lives in Africa and 7 million worldwide [1]. These estimates were generally considered low given many countries’ limited diagnostic capacity and the lack of surveillance of asymptomatic infections. [2]. Given the scale of the pandemic, caused by a novel and poorly understood virus, SARS-CoV-2 has attracted various public health response measures, including immunisation. The population can achieve immunisation of SARS-CoV-2 through natural means (becoming infected or through mother-to-child transmission) or artificial means (vaccination) [3,4,5]. However, vaccination and natural infection are more common and effective strategies than transplacental [4].

To date, global reports show low levels of vaccination in many countries. Some African countries have not yet achieved recommended vaccination levels and can be as low as 20% of the eligible population fully vaccinated [6,7,8,9]. Contributing factors include the unavailability of vaccines, religious and cultural beliefs, and vaccine myths. Nonetheless, vaccine demand remains high, and countries have created prioritisation schemes to ensure that those at the highest risk of infection and severe disease (i.e., elderly, those with comorbidities, and frontline healthcare workers) receive the vaccine first [10]. Some countries have considered administering only one dose of the 2-dose vaccine, while others have proposed that those previously infected with SARS-COV-2 should not be prioritised or could be exempted [2,11].

The global data shows that over 95% (~675 million) of those infected with SARS-CoV-2 recovered from COVID-19, meaning less than 5% died [1]. Furthermore, the seroprevalence of SARS-CoV-2 has been higher than diagnosed cases globally. In some countries, the difference has been as high as ten times the number of diagnosed cases [12,13,14], translating to more than three billion people infected and recovered globally. Many scholars also contend that given the extensive nature of the pandemic and the lack of widespread testing in most settings, it is likely that a considerable percentage of the global population has already been infected with the virus [2,13,15,16,17,18,19]. 

Given empirical evidence that infection offers long-term immunological protection against SARS-CoV-2 [2,20,21], previously infected persons could be less of a priority for vaccination. They could opt out, or their vaccination could be delayed until the supply is adequate to vaccinate the entire population [2,11,20]. Furthermore, considering previously infected individuals as immune-sensitised would help revise pandemic containment strategies and accelerate the transition back to “normal life” in most settings. 

Unfortunately, the literature on long-term immunity conferred by infection with SARS-CoV-2 is still scarce in most African countries. Only one study used a comprehensive national vast database [22]. The other is a case report in Gambia where two phylogenetically distinct SARS-CoV-2 variants reinfection cases were reported [23]. A few studies that have been conducted there did not follow up with patients long enough. Currently, global health policy and guidelines by the World Health Organization and Centres for Disease Control and Prevention (CDC) make no exceptions for persons with prior exposure to the SARS-CoV-2 infection [2,20,24]. 

Elsewhere, studies have shown low reinfection rates among those initially infected with SARS-CoV-2. The previous infection has also been shown to achieve higher herd immunity levels than some available vaccines [11,20,24,25]. Therefore, acquiring immunity through natural means, such as infection, would prove helpful for policy direction. Knowing the reinfection rate among the previously infected population would offer an insight of herd immunity levels in the general population and guide vaccination policy.

Therefore, we examined reinfection rates among the Malawian population to understand the Malawi situation using national COVID-19 surveillance data (National Line List) held at the Public Health Institute Malawi (PHIM). The primary objective was to determine the probability of testing positive for SARS-CoV-2 more than once relative to a single positive test, establish a time to reinfection, and determine the reinfection rate over time for the positive cohort. We also investigated clinical, demographic, and epidemiological risk factors for reinfection.

## 2. Materials and Methods

### 2.1. Design, Population, Setting and Study Period

This was a nation-wide retrospective cohort study including individuals (those that tested because they were identified as contacts for the COVID-19 index cases) and patients (sick and seeking health care with/without COVID-19 signs and symptoms) who tested for the COVID-19 infection via rapid test (Lateral Flow Assay (LFA), Abbott PanBio) with the diagnostic sensitivity of 98.1% (95% CI: 93.2–99.8%) and a specificity of 99.8% (95% CI: 98.6–100.0%) [25] or Reversed Transcriptase Polymerase Chain Reaction (rt-PCR, Abbott m2000) with sensitivity and specificity of 100% and 95%, respectively, in Malawi [26]. It is important to note that due to limitations of SARS-CoV-2 diagnostic resources, both LFA and rt-PCR were applied as confirmatory tests for contact index cases. The Public Health Institute of Malawi (PHIM) managed the national dataset, where all 28 districts from all five health systems and three regions reported the data. PHIM maintains a national line list for COVID-19 and other epidemic-prone communicable diseases in Malawi. We included all data from April 2020 to March 2022. The study included every person who tested for COVID-19 in Malawi during the stated period as per the Malawi National Line list held at the Public Health Institute of Malawi. These individuals had to have complete data on four instrumental variables: First name, Last name, Age and Sex. These were used to track a person through the dataset and identify reinfections. Three criteria qualify a person for a COVID-19 test in Malawi: (1) Symptomatic presentation at a health facility with symptoms suggestive of COVID-19, (2) Screening of asymptomatic travellers for issuance of COVID-19 negative certificates/travelling documents, and (3) Screening of contacts of COVID-19 index cases. We excluded all the records without variables used in this study for tracking the patients who underwent multiple COVID-19 tests. Those who did not indicate age, gender, name, and location were excluded.

### 2.2. Definition and Determination of Reinfection 

A reinfection was defined as a SARS-CoV-2-positive case on rt-PCR or rapid test (Lateral Flow Assay LFA) after at least 90 days following initial infection (rt-PCR or LFA confirmed) [27]. We ignored and never included any repeat positive tests conducted before 90 days in this study. Identification of an individual through the dataset was made based on four criteria: name, age (same age or explainable age change like where the date of birth is captured, likely within a space of 90 days; ages may change in years compared to initial age during testing), sex and phone number (where recorded). Only the first retest for which time to retest was ≥90 days was considered in this study for reinfections (i.e., multiple reinfections were not considered). Hence, individuals were followed until they became reinfected, and for those that did not become reinfected, follow-up was censored on the day the data were retrieved from the database (i.e., 10 April 2022). Therefore, this study also did not include some participants captured in the COVID-19 national list database less than three months before data retrieval since the follow-up cycle of 90 days was not reached. 

### 2.3. Data Collection Procedures

As indicated above, this retrospective study used data from all facilities that qualified to conduct COVID-19 tests across the country. These included primary healthcare facilities like health centres, secondary care facilities like district hospitals, and tertiary facilities like central hospitals. During the COVID-19 pandemic, peak points (surging of cases), the Ministry of Health had dedicated and well-trained data clerks responsible for the COVID-19 data entry, cleaning, and quality checks.

Among the data captured in the line list, demographic characteristics (age, gender, occupation, residence, etc.), clinical presentations, hospitalisation status, etc., are included. All the captured data were reported to the PHIM daily throughout COVID-19 pandemic, both during the surge (waves) and during the times when there were few cases. 

### 2.4. Data Analysis 

In this study, Public Health Institute of Malawi (PHIM) data scientists exported the data to Excel for retrieving and cleaning the data to comprehensively scrutinise the data manually and technically. For ethical purposes, the PHIM data scientists were responsible for identifying all repeated tests, regardless of whether the patients were reinfected or not. After that, they assigned a numerical anonymous identification system, deidentified the data, and handed it over to the research team. We then scrutinised the data to identify the patients who qualified to be included in the study analysis. After that, the data was exported to Stata. Data were analysed using Stata, version 13. Descriptive analyses were performed to summarise demographic and baseline clinical characteristics, time to reinfection, and reinfection rates over time. A bivariate binary logistic regression was performed to assess the association between reinfection and demographic and clinical variables. Variables demonstrating statistical significance (alpha level of 0.05) in bivariate analysis were included in a multivariable binary logistic regression to examine the independent effects of each of the variables.

## 3. Results 

As of 10 April 2022, we found 47,032 records of lateral flow assay (LFA) and rt-PCR-confirmed COVID-19 cases in the National COVID-19 database held at the Public Health of Malawi (PHIM). Upon review of the data, 817 (1.7%) records were determined to be repeat entries and were excluded from the analysis. This could likely be due to the possibility of receiving more than one test within a few days, e.g., less than a week. Further analyses were performed on the remaining 46,215 records of 45,486 individuals. From these records, the majority of the individuals tested were Malawian residents (99.37), with proportionally more males (55.76) (Table 1). From these records, 29,645 (65.17) were symptomatic, and 2156 (4.74) were hospitalised with a reported 82 (0.18) reinfection. A total of 1002 individuals were excluded from the calculation of the reinfection rate because by 10 April 2022 they had been in the cohort for less than 90 days and were therefore not eligible for assessment of reinfection. Of the remaining 44,484 individuals eligible for assessment of reinfection, 82 (0.18) were reinfected after at least 90 days of their initial infection, representing a reinfection rate of 0.55 (95% CI: 0.44–0.68) per 100,000 person-days of follow-up.

The majority of reinfections occurred in the first 90 to 200 days following the initial infection, and the median time to reinfection was 175 days (IQR: 150–314) in a range of 90–563 days (Figure 1). The risk of reinfection was highest in the immediate 3 to 6 months following the initial infection and declined substantially after that (Figure 2). The study’s age and clinical presentation demonstrated a significant association with reinfection (Table 2).

## 4. Discussion

We researched the national COVID-19 database at the Public Health of Malawi (PHIM). Just as observed in much of the published research work both locally and internationally, in this study, the majority of the individuals who tested COVID-19-positive were from local residences (Malawians) (99.37), with proportionally more males (55.76) [13,28]. The finding from this study indicates that local COVID-19 transmission was the primary driver of the pandemic as opposed to imported cases. Therefore, strategies that halt local transmission such as vaccination should be prioritised while simultaneously strengthening points of entry control measures to keep more transmissible and more virulent SARS-CoV-2 variants circulating elsewhere at bay. To date, however, vaccination rates in Malawi and many other countries, especially in Africa, are worryingly low due to both unavailability of vaccines and low uptake resulting from vaccine hesitancy caused by vaccine myths and religious and cultural beliefs [29,30]. Therefore, exploring some proven scientific approaches to help the larger population develop immunity to SAR-CoV-2 is essential. It will ensure the public health policies and guidelines are well received and suit the actual current immune status of the people. The feasible option is assessing the reinfection rate [11,21,31,32]. 

Our study found that of the 44, 484 individuals eligible for assessment of reinfection, 82 (0.18%) were reinfected after at least 90 days of their initial infection, representing a reinfection rate of 0.55 (95% CI: 0.44–0.68) per 100,000 person-days of follow-up. This finding is similar to those of other studies [22,32,33,34,35,36,37]. Although most of these studies were conducted outside the African continent, the agreement of our results and these studies confirms and re-affirms that despite geographical location or the variants most commonly indifferent and a particular area, the rate of reinfection is relatively low. It is interesting to the public health policy perspectives as it shows that most people who were once infected likely have developed immunity despite not being vaccinated. This finding fits well with the fact that seroprevalence for anti-SARS-CoV-2 has been higher than the reported numbers of COVID-19 cases in most studies, particularly in African countries, suggesting that the larger population has been exposed to the SARS-COV-2 and are likely naturally immunised [13]. However, a study conducted by Megan et al. (2021) found a relatively high reinfection rate, with protection offered from prior infection at 81.8% (95% confidence interval [CI], 76.6–85.8) [20]. These differences could be due to differences in the study coverage period under which the study was conducted. For instance, our study included participants with a follow-up period of over 23 months, while their study had a follow-up period of 6 months. However, the general outlook regarding the findings shows that the reinfection rates are commonly less than 10 per cent in most settings, which is very promising from a public health policy perspective. 

Like in other studies [20,38,39] on reinfection, most reinfections occurred in the first 90 to 200 days following initial infection, and the median time to reinfection was 175 days (IQR: 150–314), suggesting that SARS-CoV-2 protection may increase over time. A cohort study of patients with index antibody tests found that the risk of subsequent infection in individuals with positive index antibody tests decreased over time, suggesting that early positive testing represents prolonged viral shedding [39]. Like protection against many other viral infections, protection against reinfection is mediated mainly by adaptive immune memory, which has the long-term potential to maintain and reinforce pathogen-specific antibodies and effector cells [40] whereby adaptive immune responses to secondary antigen or pathogen exposures are more rapid and potent than primary responses and may substantially mitigate disease or prevent reinfection altogether, mainly via neutralising or opsonising antibodies [41]. 

Furthermore, another study assessed immunological memory in samples from COVID-19 cases and found that 95% of patients had immune memory six months after infection, including antibody or T-cell responses [20,38]. In another study, memory B cells were present for over six months, suggesting that immunity persists and may increase beyond the 150-day interval [22,32]. Therefore, it poses a potential opportunity regarding the immune status of infected people. This topic is fascinating because, like any other vaccine, all currently available WHO-approved SARS-CoV-2 vaccines do not provide complete (100%) protection against infection [42]. However, most of them, if not all, still work to reduce the severity of the infection. Particular concerns are emerging where evidence indicates that the effectiveness of almost all COVID-19 vaccines decreases as time goes on. For instance, a newly published global systematic review and meta-analysis found that, in general, almost all the COVID-19 vaccine effectiveness of the primary vaccine series against SARS-CoV-2 infections begins at an adequate level, as defined by WHO, of 83% at 14–42 days after series completion; however, vaccine effectiveness decreased significantly by 112 days after vaccination, reaching 47% by 280 days after vaccination, well below an adequate level [43]. Similarly, for COVID-19 hospitalisations and mortality, vaccine effectiveness levels were also adequate at baseline (>90%) but similarly reduced 112 days after vaccination, although vaccine effectiveness remained high over time (>75%). Practically, this poses a serious scientific demand to explore and compare the effectiveness of vaccines and natural infection on the level and longevity of protection these may offer and likely consider those already exposed to SARS-CoV-2 as an alternative legible means of being immune against COVID-19. 

Similar to other studies [21,22,23,39,40], the risk of reinfection in this study was high in the immediate 3 to 6 months following the initial infection and declined substantially after that (Figure 2), stressing the need for everyone to strictly follow COVID-19 preventive measures during this period despite previous exposure. As reported in the study by Christian et al. (2021), immunological robustness does not wane over time but becomes more robust [32]. Furthermore, research work involving multiple sample collections over a year provides evidence for long-term anti-SARS CoV-2 immunity being developed where long-term correlates of cellular and humoral immunity post-SARS-CoV-2 infection and cross-variant T cell functionality [21]. It may explain the low prevalence of reinfections and the decline of cases with time in our study. As indicated above, this is a clear difference from almost all the currently approved vaccines, hence likely being the strength of natural immunity [43]. However, one study by Snezana Medic and colleagues established that reinfection risk increases with time. In their study, the estimated risk of reinfection was 0.76%, 1.3%, 4.9%, 16.96%, and 18.86 at 6 months, 9 months, 12 months, 15 months, and 18 months, respectively. Therefore, this calls for comprehensive scientific research to explore further and compare the effectiveness of vaccines and natural infection in acquiring immunity against SARS-CoV-2 from a long-term perspective. 

Unlike other studies that revealed comorbidities, non-adherence to preventive measures, and sex as risk factors for COVID-19 reinfection [44,45,46,47], this study found none to be risk factors for COVID-19 reinfection in this cohort. However, in this study, age demonstrated an independent association with COVID-19 reinfection. Interestingly, almost all these cited studies and other research work have found age and the severity of initial COVID-19 infection to be risk factors, with older people being at a higher risk of experiencing COVID-19 reinfection and those with initial asymptomatic infection being at higher risk. This also follows the finding from a modelling study that prioritising adults aged 60+ years in vaccination remains the best way to reduce morbidity and mortality, even if the SARS-CoV-2 serostatus is known [48]. Therefore, “age” may be a critical factor to consider from a policy perspective regarding vaccination, even if patients have a prior infection with COVID-19.

A primary strength of our study is the size of our dataset, which is based on the population of Malawi and a relatively long follow-up period of nearly 23 months (April 2020 and February 2022). We took advantage of the fact that Malawi has a testing capacity, offering free testing within the population without needing a referral, regardless of age, whether an individual has symptoms or not, or whether they are suspected of being infected.

Our study is limited by a lack of access to testing results outside our health system. It might be possible and likely that some patients were tested for COVID-19 outside the health system if they had no symptoms. It is also possible that some individuals were reinfected but remained asymptomatic and did not retest, resulting in an underestimation of the reinfection rate. Nevertheless, there is little to no reason to consider that previously positive patients would be more likely to be retested outside the system than formerly negative-tested patients. Repeat positive tests could have represented persistent shedding, in which case our estimates of protective effectiveness against true reinfection are too low. Changes in behaviour following infection may have also contributed to the observed decrease in infection incidence. It is unclear whether a prior infection is associated with subsequent behavioural changes, including social distancing, mask-wearing, and test-seeking [49,50]. It is possible that following infection, individuals become more cautious, or they may feel that they have some protection, influencing the estimates [49].

## 5. Conclusions 

A comprehensive and well-elaborated understanding of the burden of SARS-CoV-2 reinfections at a national level may offer a considerable measure of a specific endurance of the immunity naturally gained and the role played by risk factors in reinfections. Such information is relevant for identifying strategies to prioritise vaccination and properly informing policymakers about the pandemic.

## Figures and Tables

**Figure 1 vaccines-11-01185-f001:**
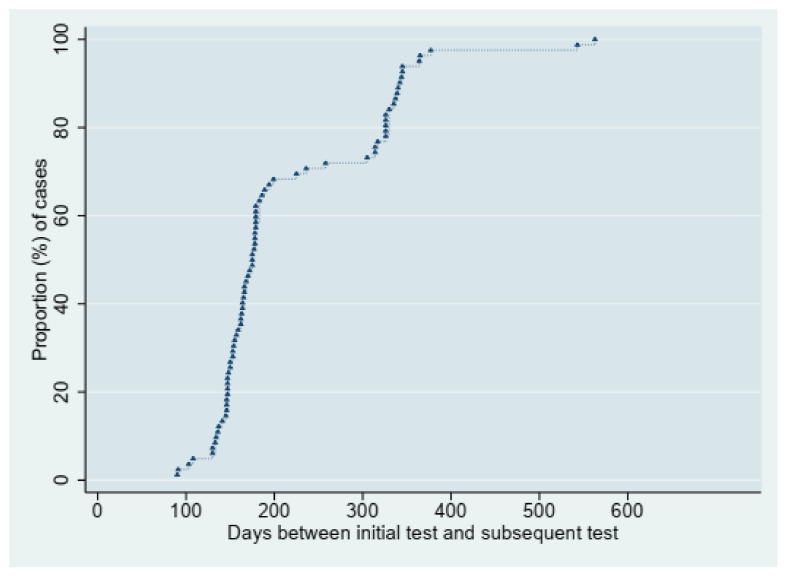
Time to reinfection for 82 reinfected individuals.

**Figure 2 vaccines-11-01185-f002:**
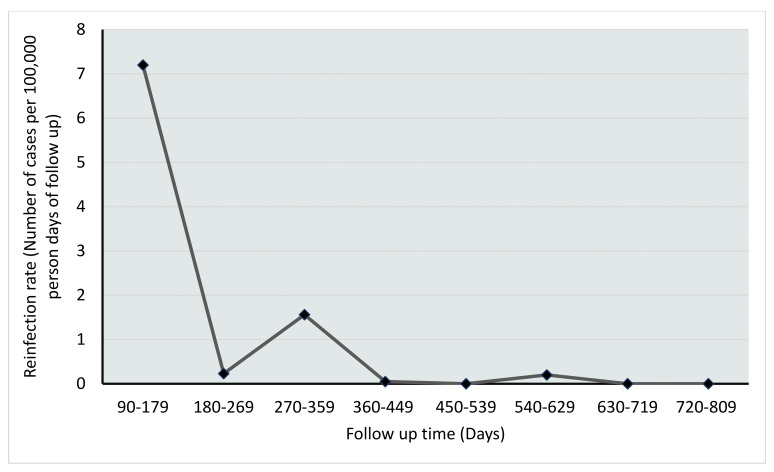
Reinfection rate over time.

**Table 1 vaccines-11-01185-t001:** Demographic Characteristics.

Variable	Frequency	Percent
Country of residence		
Malawi	43,859	99.37
South Africa	13	0.03
Other African Countries	102	0.23
Non-African Countries	165	0.37
Age		
0–19	6697	15.39
20–39	21,040	48.36
40–59	11,053	25.41
≥60	4713	10.83
Sex/Gender		
Male	25,310	55.76
Female	20,081	44.24
Presentation		
Symptomatic	29,645	65.17
Asymptomatic	11,696	25.71
Not Documented	4145	9.11
Hospitalised		
Yes	2156	4.74
No	36,595	80.45
Not documented	6735	14.81
Reinfection		
Not Reinfected	45,404	99.82
Reinfected	82	0.18

**Table 2 vaccines-11-01185-t002:** Association between reinfection and patient demographic and clinical characteristics.

Variable		Unadjusted OR (95% CI) ^†^	*p*-Value	Adjusted OR (95% CI) ^ǂ^	*p*-Value
Sex	Female (ref)	-	-	-	-
	Male	1.62 (1.02–2.56)	0.04	1.47 (0.92–2.33)	0.11
Age (years)	0–19 (ref)	-	-	-	-
	20–39	3.12 (1.24–7.84)	0.02	2.83 (1.13–7.13)	0.03
	40–59	2.79 (1.06–7.34)	0.04	2.58 (0.98–6.81)	0.06
	≥60	1.42 (0.41–4.91)	0.58	1.39 (0.40–4.84)	0.60
Clinical presentation	Asymptomatic (ref)	-	-	-	-
	Symptomatic	0.56 (0.35–0.91)	0.02	0.59 (0.36–0.95)	0.03

^†^ Bivariate binary logistic regression, ^ǂ^ Multivariate binary logistic regression, adjusted for all variables in the table.

## Data Availability

The data for the study are available from the corresponding author upon reasonable request.

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
