# Peer review of "Coronavirus Disease 2019 (COVID-19) Reinfection Rates in Malawi: A Possible Tool to Guide Vaccine Prioritisation and Immunisation Policies"

_vaccines, 2023, doi:10.3390/vaccines11071185_

Round 1

Reviewer 1 Report

This report is valuable and important to know about Malawi's pandemic situation from a humanitarian perspective and to be able to think about it together.

If authors made a find revision, it would be better for understanding.

- The highlights mean self-plagiraism or requiring re-written? It seems yes. Authors needs to revise clearly those highlighted sentences and paragraphs. 

- The materials and methods section has a wrong number; 1 instead of 2.

Highlited setences should be re-written to avoid some kinds of self-plagiarism or reproduction.

Author Response

Reviewer 1

Reviewer’s Comment - This report is valuable and important to know about Malawi's pandemic situation from a humanitarian perspective and to be able to think about it together. If authors made a find revision, it would be better for understanding.

Author’s Response: Thank you so much for kind words

Reviewer’s Comment - The highlights mean self-plagiarism or requiring re-written? It seems yes. Authors needs to revise clearly those highlighted sentences and paragraphs. 

Author’s Response: based on journal’s plagiarism checks, this paper does not contain the significant level of plagiarism threshold. I hope the journal will provide necessary explanation on the highlights

Reviewer’s Comment - The materials and methods section have a wrong number; 1 instead of 2.

Author’s Response: Thank you for your comment and correct observation. It has now been corrected. Please check.

Reviewer 2 Report

Thank you for sharing this interesting article. Here some suggested edits and comments that could help to improve the article.

L48-50 and L55-56: Content-wise possible some duplicated information; please check and revise as needed.

L64-66: The content of this sentence isn't fully clear. Does this possibly imply that ~5% did not recover? Please clarify in your manuscript.

L66-68: Not clear. Is this somehow related to diagnosed versus undiagnosed cases? Please revise your manuscript accordingly. 

L107: Please incorporate in your manuscript how the infection via contact to COVID-19 cases was confirmed? Also, did you besides positivity assess whether index cases and contact were infected by the same strain just to assure the presumed mode of transmission?

L105-110: Did you assess whether they had a single or possibly multiple infections? Did you apply any restrictions in terms of age, gender and/or past COVID-19 vaccination?

L122: How were name and phone number applied when selecting study participants? This seems rather unusual. As raised before, how were age and sex used when selecting participants? 

L118-119: Please include some information about the RT-PCR and the LFA, e.g., sensitivity and specificity, and viral sequence targeted by each test as a necessary minimal information.

L150: Please include clear inclusion and exclusion criteria of study participants. Eligibility is not fully clear up to this stage of your manuscript.

L153: Did you perform crude logistic regression or did you apply also adjustment factors to your model? If so, please explain in your manuscript including the methodological steps applied. Referring to Table 2, it seems that you did apply adjustments which must be incorporated in section 2.4 data analysis.

L156: Meaning cases were/had to be confirmed by both tests in order to be included?

L159: Please explain what you mean by "repeat entries", e.g., same case entered twice in the database, same cases tested twice. How did you differentiate repeated entries from e.g. repeated entries due to re-infection and/or relapse? 

L161: Why did you have 46215 records of just 45486 study subjects?

L168: How do you define re-infection and relapse within the context of your  manuscript? As you used RT-PCR and LFA, to what extent were you able to look into genetic information about infective strain(s) prevalent at the time of your investigation?

Table 2: Please check the journal's guidelines on how to present p-values.

L278: Did you actually look into factors such as co-morbidities and adherence to preventive measures in the context of your work?

L308: Please include a suitable reference for such a statement. 

Stated above.

Author Response

Reviewer 2

Thank you for sharing this interesting article. Here some suggested edits and comments that could help to improve the article.

Author’s Response: Thank you so much for kind words

Reviewer’s Comment - L48-50 and L55-56: Content-wise possible some duplicated information; please check and revise as needed.

Author’s response: Thank you so much for the comment and correct observation. This has been corrected by restructuring the whole paragraph.

Reviewer’s Comment - L64-66: The content of this sentence isn't fully clear. Does this possibly imply that ~5% did not recover? Please clarify in your manuscript.

Author’s response: Thank you so much for the comment and correct observation. This has been corrected. Please check

Reviewer’s Comment - L66-68: Not clear. Is this somehow related to diagnosed versus undiagnosed cases? Please revise your manuscript accordingly. 

Author’s response: Thank you so much for the comment and correct observation. This has been corrected

Reviewer’s Comment - L107: Please incorporate in your manuscript how the infection via contact to COVID-19 cases was confirmed? Also, did you besides positivity assess whether index cases and contact were infected by the same strain just to assure the presumed mode of transmission?

Author’s response: Thank you so much for the comment and correct observation. This has been incorporated. However, the data on the assessment of strain was not captured on the tool used. Malawi did not have the capacity to determine viral strains at that time.

Reviewer’s Comment - L105-110: Did you assess whether they had a single or possibly multiple infections? Did you apply any restrictions in terms of age, gender and/or past COVID-19 vaccination?

Author’s response: Thank you so much for the comment and correct observation. In this study we didn’t restrict the subject based on any of the suggested attribute and multiple infections of SARS CoV-2 assessment was not specifically done.

Reviewer’s Comment - L122: How were name and phone number applied when selecting study participants? This seems rather unusual. As raised before, how were age and sex used when selecting participants? 

Author’s response: Thank you so much for the comment and correct observation. As indicated in the manuscript, due to unavailability unique ID utilization in our health care system, there is no single variable which could be used to trace the individuals with multiple entries in the database which may entail reinfections. This is why we had to devise four variable based criteria to help identify the possible reinfection cases.   This was manually done and first and surname was the entry point of suspecting the multiple entry in the database. This was confirmed by gender, age and phone. If these four variables were similar (were in agreement) then we were sure it is the same person. We have indicated this as one of potential limitation of this study

Reviewer’s comment - L118-119: Please include some information about the RT-PCR and the LFA, e.g., sensitivity and specificity, and viral sequence targeted by each test as a necessary minimal information.

Authors response: Thank you so much for the correct observation. This has been included according to the package inserts of the manufacturing.

Reviewer’s comment - L150: Please include clear inclusion and exclusion criteria of study participants. Eligibility is not fully clear up to this stage of your manuscript.

Authors response:

Inclusion criteria: We include every person who tested for COVID-19 in Malawi during the stated period as per the Malawi National Line list held at the Public Health Institute of Malawi. These individuals had to have complete data on four instrumental variables: First name, Last name, Age and Sex. These were used to track a person through the dataset and identify reinfections. Three criteria qualify a person for a COVID-19 test in Malawi: (1) Symptomatic presentation at a health facility with symptoms suggestive of COVID-19 (2) Screening of asymptomatic travelers for issuance of COVID-19 negative certificates/travelling documents and (3) Screening of contacts of COVID-19 index cases

Exclusion criteria: We excluded entries that did not have information on either of the following: First name, Last name, Age and Sex

These have also been included the manuscript.

Reviewer’s comment - L153: Did you perform crude logistic regression or did you apply also adjustment factors to your model? If so, please explain in your manuscript including the methodological steps applied. Referring to Table 2, it seems that you did apply adjustments which must be incorporated in section 2.4 data analysis.

Authors response: We performed a bivariate logistic regression with each of the independent variables in Table 2 and then a multivariate logistic regression adjusting for all variables in Table 2. This has now been clearly stated in the methods. See data analysis section under methods

Reviewer’s comment - L156: Meaning cases were/had to be confirmed by both tests in order to be included?

Authors response: Due to limited capacity in Malawi, both were being used independently

Reviewer’s comment - L159: Please explain what you mean by "repeat entries", e.g., same case entered twice in the database, same cases tested twice. How did you differentiate repeated entries from e.g. repeated entries due to re-infection and/or relapse? 

Authors response: Repeat entries means that the same result was entered twice in the Line list (duplicate entries). Such results bore the same test date, where test date is the date that a sample was collected from an individual (depending on workload and urgency, samples could be collected today and tested a few days later).

Reviewer’s comment - L161: Why did you have 46215 records of just 45486 study subjects?

Authors response: Because some results were determined to be repeat entries (i.e., same result entered twice in the Line list) as they bore the same test date. There is a possibility however that an individual may have tested twice at two different testing sites. This may have been the case when people did not trust their LFA result and wanted a more reliable test such as rt-PCR

Reviewer’s comment - L168: How do you define re-infection and relapse within the context of your manuscript? As you used RT-PCR and LFA, to what extent were you able to look into genetic information about infective strain(s) prevalent at the time of your investigation?

Authors response: Reinfection was defined as any positive test at or after 90 days following initial infection (CDC definition). We did not try to establish whether the reinfection was a new viral strain or the persistence of the same viral strain that infected them more than 90 days ago. The country does not have the capacity to do gene sequencing, so such data is not available in Malawi.

Reviewer’s comment Table 2: Please check the journal's guidelines on how to present p-values.

Authors response: Thank you very much for the observation. This has been corrected

Reviewer’s comment - L278: Did you actually look into factors such as co-morbidities and adherence to preventive measures in the context of your work?

Authors response: Thanks so much for your comment on this. Indeed, we to look at the association between comorbidities and reinfection. But very few individuals in our data set had comorbidity and the resulting model (bivariate logistic regression) wasn’t a good fit for the data. So, we left it out. Further, the data did not include data on adherence to preventive measures.

Reviewer’s comment - L308: Please include a suitable reference for such a statement. 

 Author’s responses: Thanks so much for your comment. This has been included as per your advice.

Round 2

Reviewer 2 Report

Thank you, all my comments and suggested edits were addressed sufficiently and the manuscript has much improved. 

See above.

Author Response

Comments from the reviewer. Minor English editing is required                              Author’s Response: Thanks so much and this has been done to the best of our knowledge and ability.
